# Monitoring Roadbed Stability in Permafrost Area of Qinghai–Tibet Railway by MT-InSAR Technology

Hui Liu [1,2], Songbo Huang [1,2], Chou Xie [3,4,*], Bangsen Tian [3,4,*], Mi Chen [1,2] and Zhanqiang Chang [1,2]

1 College of Resources Environment and Tourism, Capital Normal University, Beijing 100048, China
2 Key Lab of 3D Information Acquisition and Application, Capital Normal University, Beijing 100048, China
3 Aerospace Information Research Institute, Chinese Academy of Sciences, Beijing 100094, China
4 University of Chinese Academy of Sciences, Beijing 100049, China
* Correspondence: xiechou@aircas.ac.cn (C.X.); tianbs@aircas.ac.cn (B.T.)

**Abstract:** Permafrost areas pose a threat to the safe operation of linear projects such as the Qinghai–Tibet railway due to the repeated alternating effects of frost heaving and thawing settlement of frozen soil in permafrost area. Time series InSAR technology can effectively obtain ground deformation information with an accuracy of up to millimeters. Therefore, it is of great significance to use time series InSAR technology to monitor the deformation of the permafrost section of the Qinghai–Tibet railway. This study uses multi-time InSAR (MT-InSAR) technology to monitor the deformation of the whole section of the Qinghai–Tibet railway, detect the uneven settlement of the railway roadbed in space, and detect the seasonal changes in the roadbed in the time domain. At the same time, the local deformation sections over the years are compared and discussed. The time series deformation monitoring results of the permafrost section Sentinel-1 data in 2020 show that the length of the railway roadbed from Tanggula station to Za'gya Zangbo station (TZ) section is approximately 620 m, the deformation of the east and west sides is uneven, and the average annual deformation difference is 60.68 mm/a. The impact of frozen soil in WangKun station to Budongquan station (WB) section on railway roadbed shows the distribution characteristics of high in the middle and low at both ends, and the maximum annual average settlement can reach −158.46 mm/a. This study shows that the deformation of permafrost varies with different ground layers. The impact of human activities on frozen soil deformation is less than that of topography and hydrothermal conditions. At the same time, the study determined that compared with other sections, the roadbed deformation of TZ and WB sections is more obvious.

**Keywords:** MT-InSAR; frozen soil deformation; Qinghai–Tibet corridor; roadbed deformation; climate response





## 1. Introduction

The repeated alternating action of frost heaving and thawing settlement of frozen soil in permafrost area will cause damage to the local geological environment, and then easily lead to geological disasters such as landslide, debris flow, foundation rupture, and collapse [1,2], especially linear engineering such as railway and highway [3–5]. Uneven deformation of the roadbed is a common and serious disaster in linear engineering, which seriously affects the operation of linear engineering such as high-speed railways and highways. The Qinghai–Tibet Plateau (QTP) is the largest permafrost region except polar regions [6]. Approximately 610 km of Qinghai–Tibet railway is laid in the permafrost area, crossing national nature reserves such as Hoh Xil, Sanjiangyuan, Qiangtang. In view of the particularity and importance of the location of the Qinghai–Tibet railway, it is of great practical significance to monitor the surface deformation of the Qinghai–Tibet corridor in permafrost area [7–17].

Traditional geodetic methods, such as leveling and global positioning system (GPS) measurement, can achieve high-precision measurement of surface deformation. However,

due to the special location and harsh environmental conditions of the Qinghai–Tibet plateau, both traditional leveling and GPS measurement require a lot of human and material costs, which is difficult and inefficient. Although it will not cause this problem to predict the stability of Qinghai–Tibet line foundation in combination with historical disaster data, field survey data [18], settlement index or allowable bearing capacity index [19], the accuracy is rough and low, and the prediction of influence factors cannot be better applicable to the existing situation under the external human intervention.

The wide application of synthetic aperture radar interferometry (InSAR) makes up for the shortcomings of the above method. InSAR monitors ground deformation by analyzing the phase information of two aperture radar images [20]. It has been widely used in the ground deformation caused by earthquakes, volcanoes, glaciers, landslides, and land subsidence [21–24]. With the launch of new SAR satellites such as Sentinel-1A/B, researchers can more easily obtain SAR images. Rich SAR data sources and accessibility promote the application of InSAR technology in the field of permafrost deformation monitoring. Due to its large coverage of SAR images and the relatively short return visit time, the interference coherence has been greatly improved, which better meets the accuracy requirements of deformation monitoring such as linear engineering in permafrost area, and it is used to monitor frozen soil deformation. Researchers have already demonstrated the ability to use InSAR technique to detect to this freeze/thaw-related ground motion over permafrost regions. As shown in Table 1, current researches using InSAR technology to monitor frozen soil mainly focus on the Arctic, Qinghai–Tibet Plateau and other regions of the world. It can be seen that the application of InSAR technology to permafrost monitoring has broad prospects.

**Table 1.** Research on monitoring frozen soil with InSAR technology.

| Arctic | Qinghai–Tibet Plateau | Other Regions |
| --- | --- | --- |
| Zwieback et al. [25], Bartsch et al. [26], Strozzi et al. [27], Rudy et al. [28], Liu et al. [29–33] | Zhou et al. [34], Xu et al. [35], Zou et al. [36], Xiang et al. [37], Zhang et al. [38–44], Wang et al. [45,46], Reinosch et al. [47], Lu et al. [48], Chen et al. [49], Daout et al. [50] | Chen et al. [51], Rouyet et al. [52], Antonova et al. [53], Li et al. [54], Liu et al. [29] |

As can be seen from the above table, the research on permafrost in the Qinghai–Tibet plateau in recent years has been fruitful. However, over the years, most research on the frozen soil deformation along the Qinghai–Tibet railway has focused on the section from Wudaoliang to Tuotuohe, especially the Beiluhe area, and less so on the Chaerhan Salt Lake section in the north and the section from Yangbajing to Dangxiong in the south. The scope of the existing research has been very limited, and the overall deformation of the permafrost section of the Qinghai–Tibet railway has not been monitored. In addition, the existing research only involves local disaster characteristics, and there is a lack of systematic investigation and analysis of the overall disaster characteristics and laws. Most of the deformation research on the frozen soil section of the Qinghai–Tibet railway is limited to the results of more than a decade. Due to the long time, it is difficult to reflect the current situation of roadbed deformation, which is not conductive to the discovery of the existence and potential risk of deformation of railway roadbed. At the same time, the previous research results have not been compared and discussed, and the methods and effects of the measures have not been qualitatively and quantitatively evaluated. The relevant research only stays at the level of producing deformation results. Therefore, many years after the completion of the Qinghai–Tibet railway project, it is very necessary to

compare the deformation monitoring results in recent years with the previous monitoring results to understand whether the deformation disaster has changed.

Most of the studies have only used permanent scatterers to obtain the deformation characteristics of ground objects along the Qinghai–Tibet railway. The scattering characteristics are directly affected by external environmental factors such as soil moisture and surface water, and almost no ground objects can maintain stable backscattering characteristics for a long time. At the same time, the vegetation coverage in the study area is low, mostly exposed sandy or mucus saline soil, and the interference coherence is high in the short term. This phenomenon makes it more difficult to obtain persistent scattering points. Therefore, it is difficult to obtain high-density stable scattering points only by obtaining traditional persistent scatterer (PS) points. Distributed scatterers (DS) refers to ground objects, mostly bare ground and sparse vegetation that resolve the backscattering coefficients of all scatterers in a unit roughly the same. The spatial density of the measurement points is increased on the region characterized by DS while retaining the high-quality information obtained using the PS technique on deterministic targets. PS and DS were properly combined to increase the density of measurement points and further improve the coherence point density and parameter estimation accuracy [55]. Considering the limitations of traditional persistent scattering points, we obtain a high-quality and high-density time series deformation point set by combining PS points and distributed scattering points (DS) with limited time baselines, which greatly improves the defects of the above methods so as to obtain the existing deformation of railway roadbed and potential risks along the route.

In this study, we used the free obtained Sentinel-1A Radar image to monitor the overall deformation of the permafrost section of the Qinghai–Tibet railway by combining PS points and DS points, compared and discussed the deformation of local areas for many years to monitor the uneven settlement of railway roadbed in space and the seasonal changes in the time domain.

## 2. Materials and Methods

### 2.1. Study Area

The repeated alternating action of frost heaving and thawing settlement of frozen soil in permafrost area will have a serious impact on linear projects such as Qinghai–Tibet railway. Approximately 610 km of Qinghai–Tibet railway is paved in permafrost area, from Nachitai to Ando station. The study area covers 32°20′–35°94′, 91°42′–94°62′, with an altitude of 3514–5544 m.

The thickness of frozen soil is approximately 60~120 m, and the thickness of active layer (ALT) is between 0.8 and 4 m, with an average of approximately 2 m It is a typical continental climate. It is cold in winter (to −20.7 °C), warm in summer (to +22.7 °C), and the annual average temperature is −0.98–1.03 °C. The ground is generally frozen for 7 months (October to April of the next year) [56]. The annual precipitation is between 191 and 485 mm, mainly concentrated in summer (May to October). The elevation fluctuates from north to south [57], and its topographic characteristics are shown in Figure 1. There are four monitoring stations close to this section, namely Golmud, Wudaoliang, Tuotuohe and Anduo meteorological stations. Land types mainly include glaciers, snow, exposed rocks and other land types [58]. Due to the melting and freezing of frozen soil, the maximum seasonal deformation of the surface can reach 10 cm. The distribution of frozen soil is shown in the upper left corner of Figure 1. The landslide within 1 km and debris flow within 5 km caused by frost heaving and thawing settlement of frozen soil will pose a threat to the railway. This study will take the permafrost section of a 5 km buffer zone as the research object.

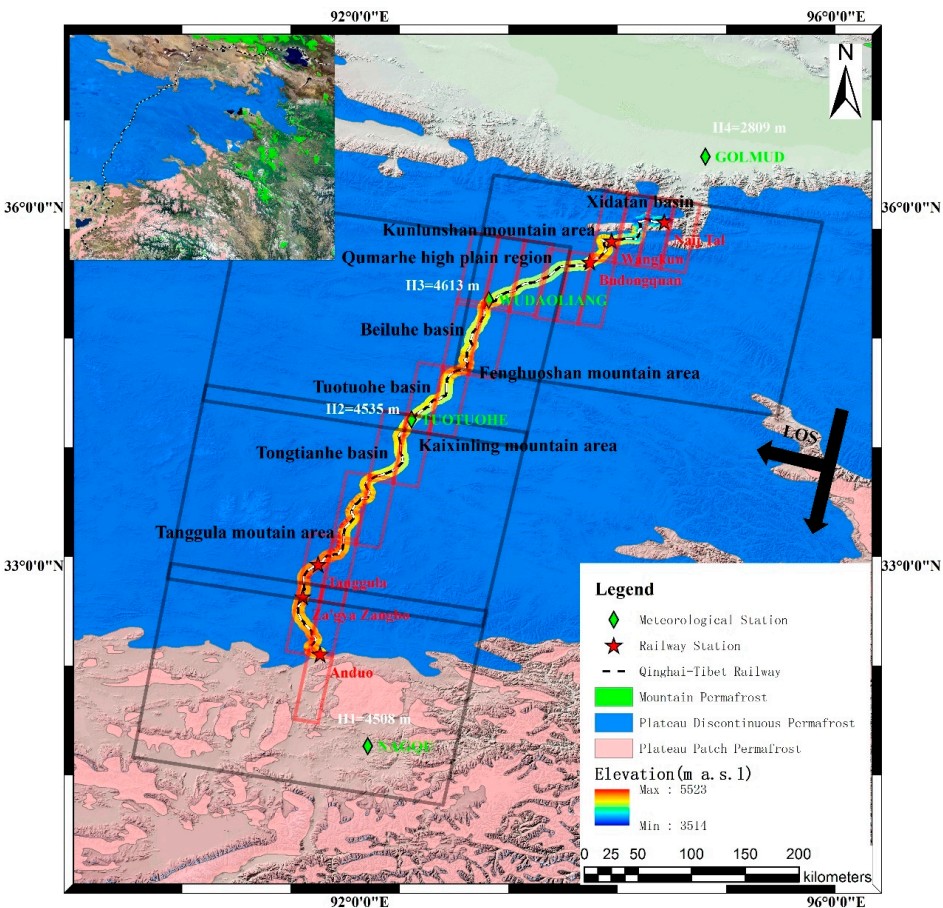

**Figure 1.** The green diamond is the meteorological station. The station name is marked in green font. The white mark is the altitude of each meteorological station. The red pentagram is the railway station along the way, and the name is marked in red font. The terrain distribution has been marked in black font in the Figure. The Qinghai–Tibet line passes through mountain permafrost, plateau discontinuous permafrost and island permafrost from north to south. The study area is a permafrost section passing through the blue area, which has higher risk than other sections. The black rectangular box indicates the coverage of the Sentinel-1 image, the red rectangular box is the swath of each scene used in the study, and the black arrow indicates the navigation direction and line of sight direction.

### 2.2. Data

In this paper, the descending image of Sentinel-1 satellite in 2020 is selected to obtain the risk situation of the section of Qinghai–Tibet line, involving 122 images in four maps, as shown in Table 2.

**Table 2.** Relevant parameters of four scenes of radar image.

| Sequence | Start Time | End Time | Path/Frame | Image | Swath |
|---|---|---|---|---|---|
| 1 | 12 January 2020 | 25 December 2020 | 77/475 | 29 | 7 |
| 2 | 5 January 2020 | 30 December 2020 | 150/475 | 31 | 7 |
| 3 | 5 January 2020 | 30 December 2020 | 150/480 | 31 | 6 |
| 4 | 5 January 2020 | 30 December 2020 | 150/485 | 31 | 3 |

The pixel sizes in the central azimuth and range directions of the image are 2.32 m and 13.98 m, respectively, and VV polarization mode is selected. The central incidence angle is between 33 and 34°, and the spatial range covered by each SAR image is approximately $252 \times 190$ km$^2$. The digital elevation model (DEM) of the study area adopts SRTM-1 with a resolution of 30 m. In addition, the Precision Orbit Data (POD) of Sentinel-1A is provided

by European Space Agency (ESA). The data of glacier and frozen soil distribution originate from the "Heihe project data management center" [59].

*2.3. Methods*

In this study, permanent scatterers and distributed scatterers (PS, DS) were combined to select highly coherent targets and increase the coverage of coherent points. Compared with PS method, this method has a better effect and higher efficiency in the analysis of foundation deformation along the Qinghai–Tibet corridor. The processing flow chart is demonstrated in Figure 2. The deformation results largely depend on the error correction during interference processing, the selection of coherent points and phase unwrapping. Therefore, this section is expanded with the principles and methods of each part to finally obtain the desired results.

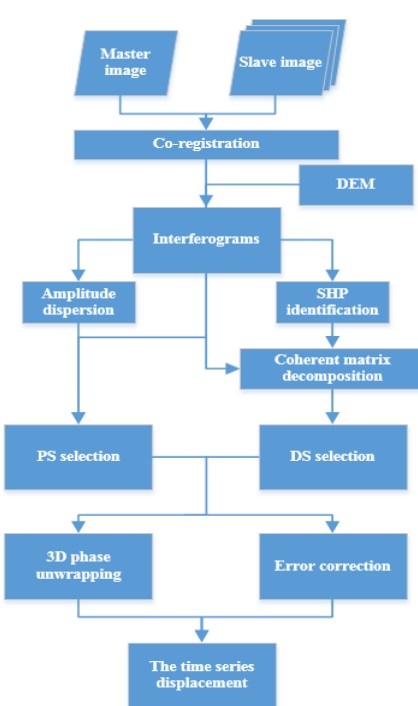

**Figure 2.** Process of joint point selection of PS and DS to obtain high-density point targets and roadbed time series deformation.

2.3.1. Error Correction

High coherence of target means high coherence in space domain and stable phase in time domain. For MT-InSAR, the image will be registered, and the terrain and orbit have generated phase compensation. The residual phase $\varphi^n$ of unwrapped interferogram obtained after differential interference is usually composed of residual terrain, deformation, atmosphere and noise phases [60], which are wound between deformation phases.

$$\varphi^n = \varphi^n_{topo} + \varphi^n_{defo} + \varphi^n_{aps} + \varphi^n_{noise} + k \cdot 2\pi, \tag{1}$$

where $\varphi^n_{topo}$ is the residual topographic phase; $\varphi^n_{defo}$ is the deformation phase; $\varphi^n_{aps}$ is the atmospheric phase screen, indicating the signal delay caused by weather conditions; $k$ is the integer fuzzy number; $\varphi^n_{noise}$ is the phase noise caused by time decorrelation, error registration, uncompensated spectral offset decorrelation, orbit error, soil moisture and thermal noise.

The residual topographic phase of point p is as follows:

$$\varphi^n_{p,topo} = k^z_p \cdot \epsilon^z_{p}, \tag{2}$$

where $k_p^z = \frac{4\pi b_n}{\lambda sin\theta R_n}$ is height-phase coefficient and $\epsilon_p^z$ is elevation error; $b_n$ is the baseline of the nth image relative to the reference (main) image; $\theta$ is the local angle of incidence; $\lambda$ is the carrier wavelength; $R_n$ is the (zero Doppler) distance between the target and the nth orbit acquisition.

The deformation phase generated by the displacement at point p can be effectively divided into two components:

$$\varphi_{p,defo}^n = k^t v_p + \mu_{NL},\tag{3}$$

where $v$ is constant speed (line of sight); $k^t = \frac{4\pi t_n}{\lambda}$ is the time phase factor; $t_n$ is the time baseline; $\mu_{NL}$ is the phase term due to possible nonlinear motion.

Atmospheric phase screen (APS) can greatly reduce the influence by considering the phase difference between two nearby points. Therefore, APS calibration can also be carried out by establishing Delaunay triangulation network.

For the two PS candidate points connected, the measured tomography signals $y_1(m)$ and $y_2(m)$ can be expressed as [61]

$$\begin{cases} y_1(m) = g_1(m, s_1) exp[j\varnothing_{APS1}(m)] \\ y_2(m) = g_2(m, s_2) exp[j\varnothing_{APS2}(m)] \end{cases},\tag{4}$$

where $\varnothing_{APS1}(m)$ and $\varnothing_{APS2}(m)$ represent APS present in $y_1(m)$ and $y_2(m)$, respectively; $s_1$ and $s_2$ are the inclined elevations of the two PS points connected, respectively; $g_1(m, s_1)$ and $g_2(m, s_2)$ represent the math ideal measurement value of the two connection points without APS interference.

Because the spatial frequency of APS is low, the candidate points with long spatial distance usually have great differences in APS. At this time, it is difficult to deal with connection points with large APS differences. Therefore, the long connection arc is rejected by setting the distance threshold. When the arc length is short, two adjacent scatterer candidate points have similar APS:

$$\varnothing_{APS1}(m) \approx \varnothing_{APS2}(m).\tag{5}$$

Therefore, the APS can be easily calibrated by subtracting the phase of the reference point, and the relative tomographic signal $\Delta y(m)$ of the connecting arc is as follows:

$$\Delta y(m) = y_2(m) exp(-j \cdot \lceil y_1(m) \rceil) = g_2(m, s_2) \cdot exp(-j \cdot \lceil g_1(m, s_1) \rceil),\tag{6}$$

where $\lceil \ \rceil$ is the operation of phase retention.

2.3.2. Selection of Coherent Points

Permanent and distributed scatterers are combined to increase the coverage of coherent points. Ferretti et al. proposed the Dispersion of Amplitude (DA) based on the definition of PS [62]. When the main scatterer exists in the pixel, its phase is mainly determined by the phase of the main scatterer, which is less affected by noise, and the phase standard deviation has the following relationship with the amplitude:

$$\sigma_\varphi \approx \frac{\sigma_A}{m_A} \equiv D_A,\tag{7}$$

where $\sigma_\varphi$ is the standard deviation of phase; $\sigma_A$ is the standard deviation of amplitude A; $m_A$ is the mean amplitude of N SAR images in time dimension; $D_A$ is the dispersion index.

When extracting DS, considering its statistical distribution characteristics [63], the extracted DS is transformed into extracting several pixels subject to the same statistical distribution, which can be called homogeneous pixels. Then, the recognition of planar targets is transformed into identifying homogeneous pixels first, and then estimating their phase. The Kolmogorov–Smirnov algorithm is used to identify homogeneous pixels. By

determining whether their cumulative distribution function (CDF) is the same, we can judge whether they are homogeneous pixels. K–S algorithm defines $D_N$ as the maximum absolute value of the difference between the two cumulative distribution functions:

$$D_N = \max_{-\infty < x < \infty} |S_N(x) - P_N(x)|, \tag{8}$$

where $S_N$ and $P_N$ are the cumulative distribution functions of two different SAR image pixels, respectively. By setting a certain threshold ($D_{thr}$), when $D_N \leq D_{thr}$, it is considered that the two samples obey the same statistical distribution, that is, they are judged to be homogeneous pixels so as to ensure that the identified homogeneous pixel is directly or indirectly connected with the central pixel P, and there is no independent region. In addition, the algorithm takes the identified homogeneous pixel as DS, which makes the subsequent phase estimation more accurate, so the results are more reliable.

### 2.3.3. Phase Unwrapping

The accuracy of surface deformation information acquisition mainly depends on phase unwrapping. The main idea of phase unwrapping algorithm based on network flow is to minimize the difference between the derivative of unwrapping phase and the derivative of winding phase. This method can not only greatly reduce the time and space complexity of phase unwrapping algorithm and improve the calculation speed, but also limit the whole error to a small range and prevent the retransmission of error, so as to improve the accuracy of unwrapping results. In this study, the minimum cost flow algorithm is based on irregular networks: first, the phase with high coherence coefficient is extracted as a high-quality phase data set. Next, a Delaunay triangulation is established according to the position of these phase points. Then, the residual points in the triangulation are identified, and the minimum cost flow algorithm is used to connect the positive and negative residual point pairs to establish the branch tangent. Finally, the unwrapping phase value is obtained by the method of adding and subtracting $2n\pi$ around or through the branch tangent.

The phase difference between two adjacent phase points (such as targets p and q) can be written as [64]

$$\Delta\varphi_{pq}^n = W\left\{ k_p^z \Delta\epsilon_{pq}^z + k^t \Delta v_{pq} + \Delta w_{pq}^n \right\}, \tag{9}$$

where $W$ is the winding operator, $\Delta\epsilon_{pq}^z$ is the relative elevation error, $\Delta v_{pq}$ is the relatively constant velocity, and $\Delta w_{pq}^n = \mu_{pq,NL}^n + \varphi_{pq,aps}^n + \varphi_{pq,noise}^n$ is the phase difference between the model and measured values between p and q points in the nth interferogram.

Therefore, the Delaunay triangulation connection network is constructed in the spatial domain as follows. Assuming that the largest connected network is searched after K iterations, and there are $P^{(n)}$ arcs and $Q^{(n)}$ PS points (n = 1, $\cdots$, K) in the nth search result of the connected network, then the maximum connected network contains $P^{(k)}$ connected arcs and $Q^{(k)}$ points. The integration of the nth search connection network can be modeled as

$$\Delta S^{(n)} = G^{(n)} \cdot S^{(n)}, \tag{10}$$

where $\Delta S^{(n)}$ is the relative height of the $P^{(n)}$ point arc of the nth search connection network; $G^{(n)}$ is the transformation matrix from point arc to point, which is composed of $0, -1, 1$; $S^{(n)}$ is the absolute height of the $Q^{(n)}$ point in the nth search connection network:

$$\Delta S^{(n)} = \begin{bmatrix} \Delta \hat{S}_1^{(n)} \\ \Delta \hat{S}_2^{(n)} \\ \vdots \\ \Delta \hat{S}_p^{(n)} \\ p \end{bmatrix}_p^{(n)} \times 1, G^{(n)} = \begin{bmatrix} 1 & \cdots & -1 & \cdots & 0 & \cdots \\ \vdots & & & & & \\ \cdots & 1 & 0 & \cdots & -1 & \cdots \\ \vdots & & & & & \\ \cdots & 0 & \cdots & 1 & -1 & \cdots \\ \vdots & & & & & \end{bmatrix}_{p^{(n)} \times Q^{(n)}}, S^{(n)} = \begin{bmatrix} \Delta S_1^{(n)} \\ \Delta S_2^{(n)} \\ \vdots \\ \Delta S_{Q(n)}^{(n)} \end{bmatrix}_{Q^{(n)} \times 1}. \quad (11)$$

## 3. Results

In order to improve the phase solution accuracy, it is necessary to check whether the spatio-temporal baseline meets the requirements, and reduce the impact on the deformation phase. In the study area, 21 swaths were analyzed by MT-InSAR, and a total of 118 pairs of interference pairs were generated. The temporal and spatial baseline distribution and combination mode are shown in Figure 3. The re-entry period of SAR satellite is 12 days, the vertical baseline is less than 130 m, and the coherence is relatively high. The finally obtained surface time series deformation information along the Qinghai–Tibet line is shown in Figure 4. The areas with serious deformation are mainly distributed from WangKun station to Budongquan station (WB) in the north section of permafrost area and from Tanggula station to Za'gya Zangbo station (TZ) in the south section.

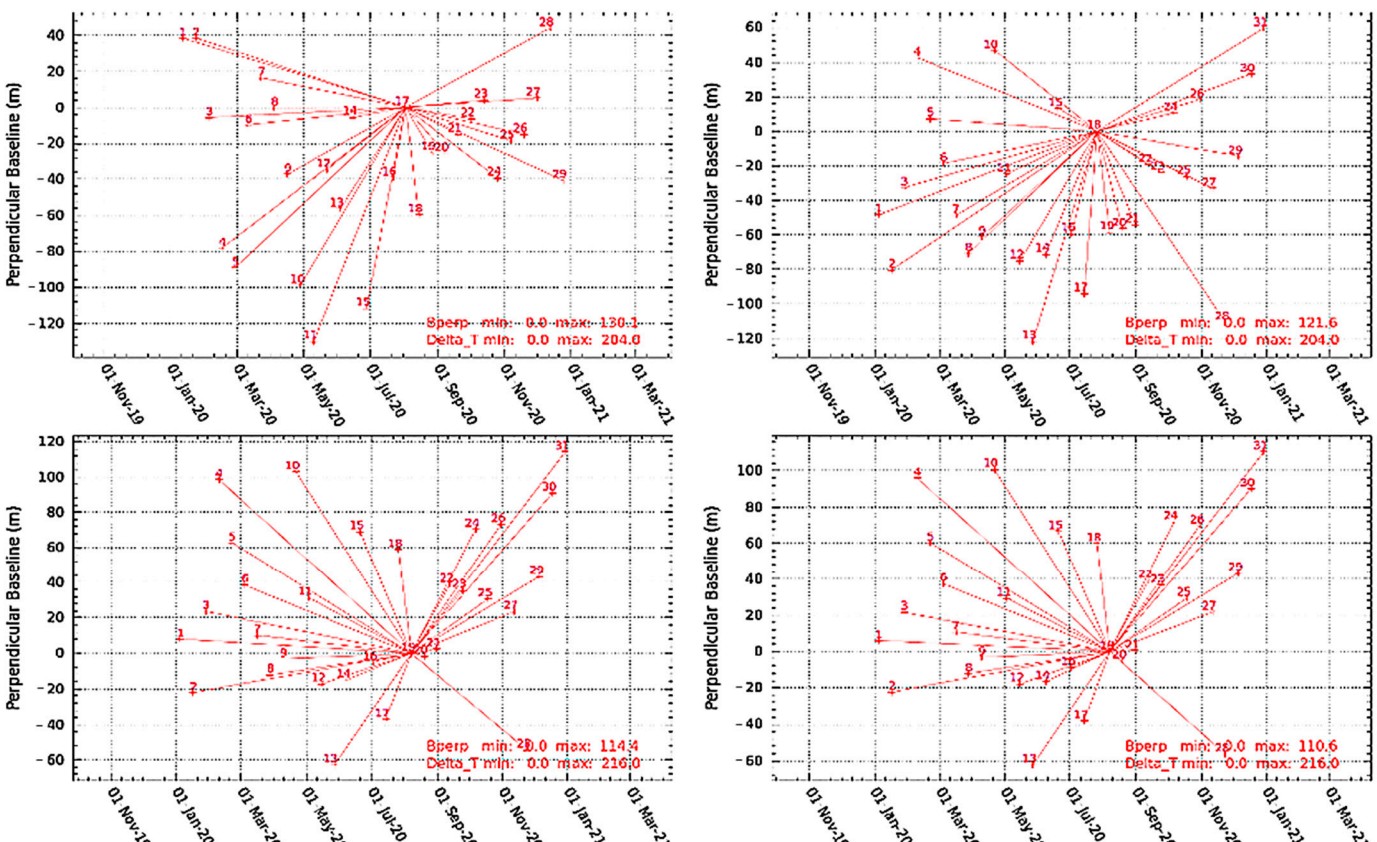

**Figure 3.** The center point is the reference main image, each line represents an interference pair, and the X and Y axes represent its spatio-temporal baseline distribution.

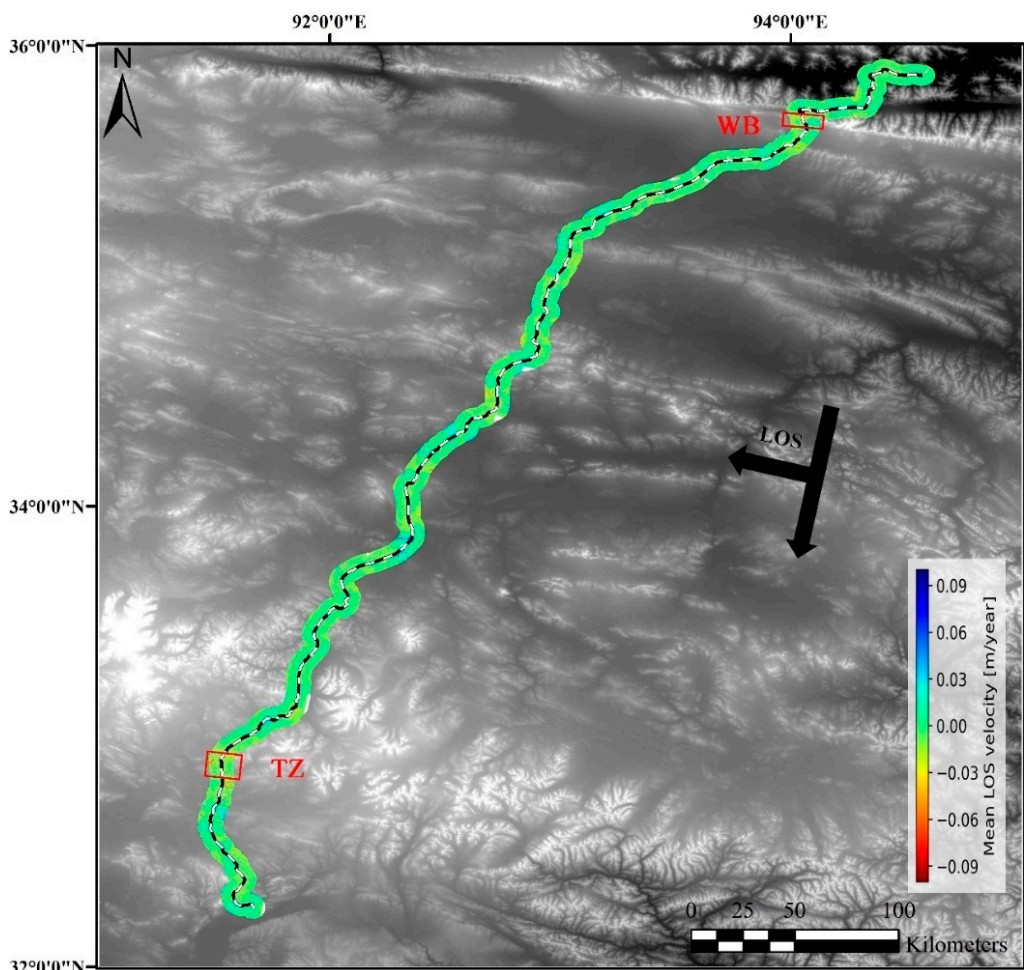

**Figure 4.** Overall deformation results obtained by time series InSAR processing. The surface deformation of frozen soil section of Qinghai–Tibet corridor is mainly distributed in WB in the north and TZ in the south, which is represented by red rectangular box in the Figure. The black arrow indicates the Los direction.

## 4. Discussion

### 4.1. Risk Section and Deformation Law

Affected by the thermal thawing and frost heaving of the active layer of permafrost, there are roadbed deformation and frozen soil collapse in WB and TZ of the two sections, as shown in Figures 5a and 5a', respectively, representing the original amplitude map and annual average deformation of WB section in 2020. The deformation is mostly concentrated in the valley area in the north and appears to be lesser in the flat area in the south. Figure 5b,b' is the original amplitude map and annual average deformation map of TZ section. As it passes through the inland river, it is greatly affected by runoff and thermal melting, resulting in melting collapse in many places in the southern section of the area.

#### 4.1.1. Deformation from WangKun Station to Budongquan Station

The deformation section between WangKun station and Budongquan station is located in the valley area, and its terrain, deformation and geological structure, is shown in Figure 6.

Affected by global warming and human activities, the phenomenon of frost heaving and thawing settlement of permafrost active layer in this area is obvious. Because the terrain of this section is high on both sides and low in the middle, when the shear force reaches a certain threshold due to the thawing settlement of frozen soil, a shallow landslide will occur at the front of the mountains on both sides. At the same time, because this section is located in the valley, there is a large amount of glacier melt water and precipitation, and the surface

and underground runoff is more abundant. The hydrothermal effect will continuously thicken the frozen soil active layer, aggravate the frost heave and thaw settlement, and further promote the collapse on both sides of this section. It can also be clearly seen in Figure 6d that melting collapse occurs in R1 and R2 areas along both sides of the railway section. The center latitude and longitude of R1 is 35.66984° N, 94.05412° E; the average annual deformation rate is −62.91 mm/a; and the collapse area reaches 0.36 km². The deformation on both sides of the railway at R2 is uneven, with roadbed lifting on the left and thawing collapse of frozen soil on the right. The annual deformation is 48.13 mm/a and −158.46 mm/a, respectively. The existence of large-scale melt collapse body and the difference in roadbed deformation affect the normal operation of railway. In addition, since the railway section in flat areas is basically free of deformation, it can be inferred that the influence of human activities of railway operation on frozen soil deformation is smaller than that of topography and hydrothermal. In other words, topography and hydrothermal action are the main causes of railway deformation in this section.

Considering that geological factors may also affect it, we compared the geological distribution Figure 6c with the deformation distribution Figure 6d. It was determined that the surface deformation of this section is mainly concentrated between the two faults, and the deformation is weak in the north of the North fault. At the same time, there are Qp2gl, Qp3gl, TB3, P1QQ3 and other strata between the two fault layers. The severely deformed strata are mainly middle Pleistocene ice deposit Qp2gl and late Pleistocene ice deposit Qp3gl, which are mainly ice water accumulation. The moraine is composed of boulders, gravel, sand and clay. The soil is soft, sensitive to hydrothermal changes, and prone to hot melting and frost heaving. Therefore, the roadbed deformation in frozen soil section may be affected by thawing settlement of frozen soil, fault, and lithology.

Due to the complex physical movement along the slope of this section, more information (movement rate, direction, and trend) about terrain (gradient and direction), geology (mantle composition and surface coverage), hydrology (surface and underground runoff and ice melting of permafrost) is needed to monitor the evolution of active layer of permafrost so as to better grasp the deformation inducement and law of this area.

4.1.2. Deformation from Tanggula Station to Za'gya Zangbo Station

The section with serious roadbed deformation from Tanggula station to Zajiazangbu station is selected. The whole section shows a downward trend, with uneven deformation on both east and west sides. The total length of the section is approximately 620 m, and the central latitude and longitude are 32.90665° N, 91.52807° E. This section passes through Za'gya Zangbo, the longest inland river in Tibet. The runoff is supplemented by the water melted by ice and snow. As many sections of runoff pass through this section, the river has a great impact on the thermal thawing of permafrost, resulting in changes in the hydrothermal status of the active layer above the permafrost [65]. Coupled with the joint impact of railway operation activities, the phenomenon of frost heaving and thawing settlement of the frozen soil layer is obvious.

The red arrow shows the connection between the north and south sections. It can be observed that the deformation of the south section is larger than that of the north section as a whole. In Figure 7a, the annual average deformation phase at P3 is smaller than that of P1 and P2. The water activity and temperature distribution of railway roadbed in frozen soil area are the key factors affecting roadbed frost heaving and thawing settlement. In unsaturated state, the higher the water content, the greater the frost heaving amount of soil with the same density, the greater the corresponding frost heaving and thawing settlement rate; that is, water supply is the fundamental reason for frost heaving and thawing settlement of frozen soil [6,66]. There are two branches of runoff in the south, which have a connecting trend, and the runoff at the arrow is expanding, which has a great impact on the frozen soil. The frost heaving and thawing settlement process of frozen soil active layer is more intense, resulting in uneven roadbed deformation on the north and south sides.

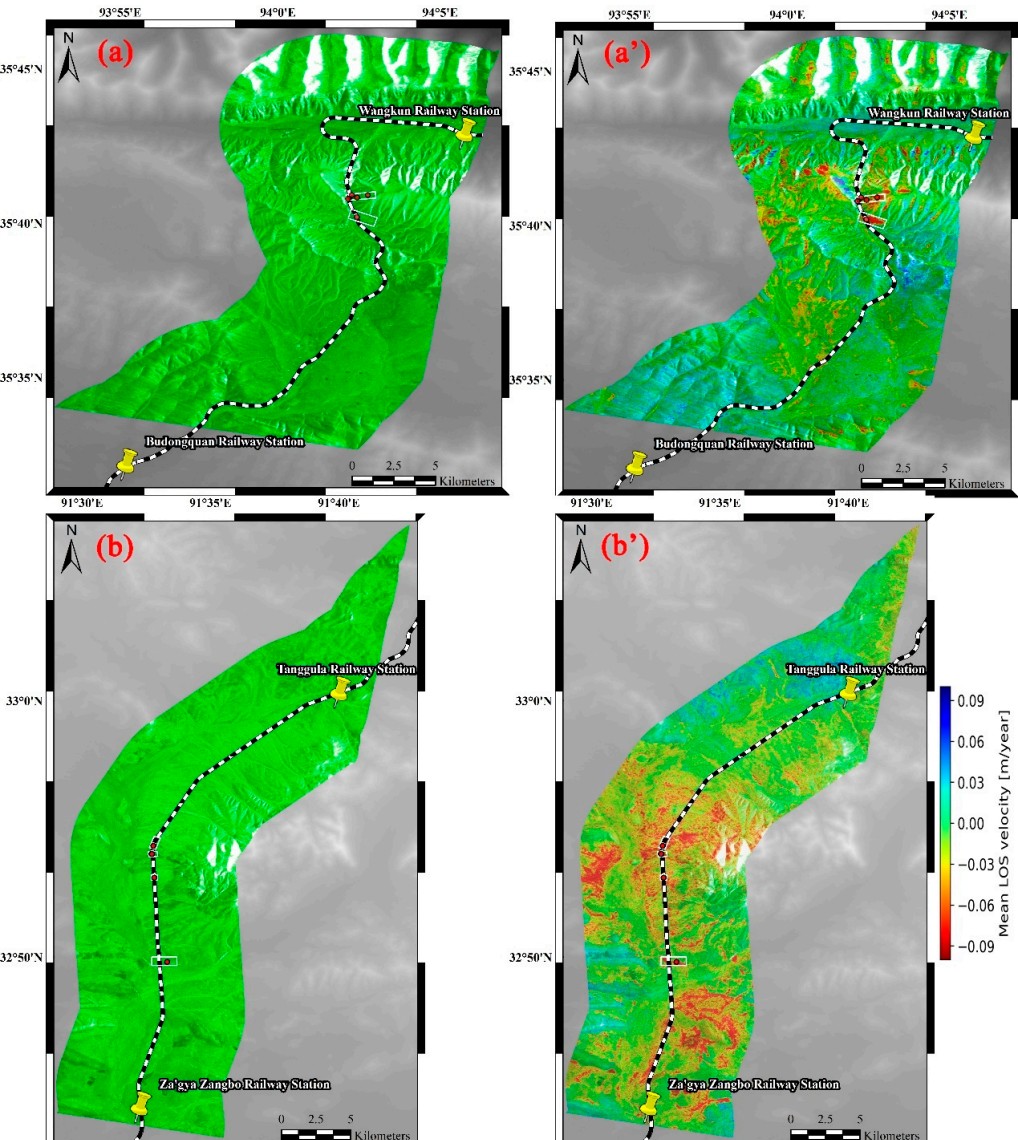

**Figure 5.** (**a**,**a'**) are the original amplitude map and annual average deformation map of WB deformation section, respectively (the annual average deformation of the surface after being affected by the thermal thawing and frost heaving of permafrost active layer), (**b**,**b'**) are the original amplitude map and annual average deformation map of TZ deformation section, respectively, In the Figure, the red solid dot indicates the roadbed deformation or serious deformation area threatening the roadbed, and the white box indicates the deformation risk section.

In the south section, the deformation on the left and right sides is uneven and the difference is obvious, which poses a threat to the railway operation. As shown in P1 and P2 in Figure 7c, the cumulative deformation is 113.54 mm and 54.76 mm, respectively, and the annual average deformation is −112.768 mm/a and −52.084 mm/a, respectively. The uneven deformation may be affected by the thickness difference in the active layer of permafrost. The thermal melting of frozen soil in summer leads to different collapse degrees on both sides of the railway, and it cannot be completely frozen in winter, resulting in greater and greater differences on both sides of the roadbed and affecting the stability of the roadbed.

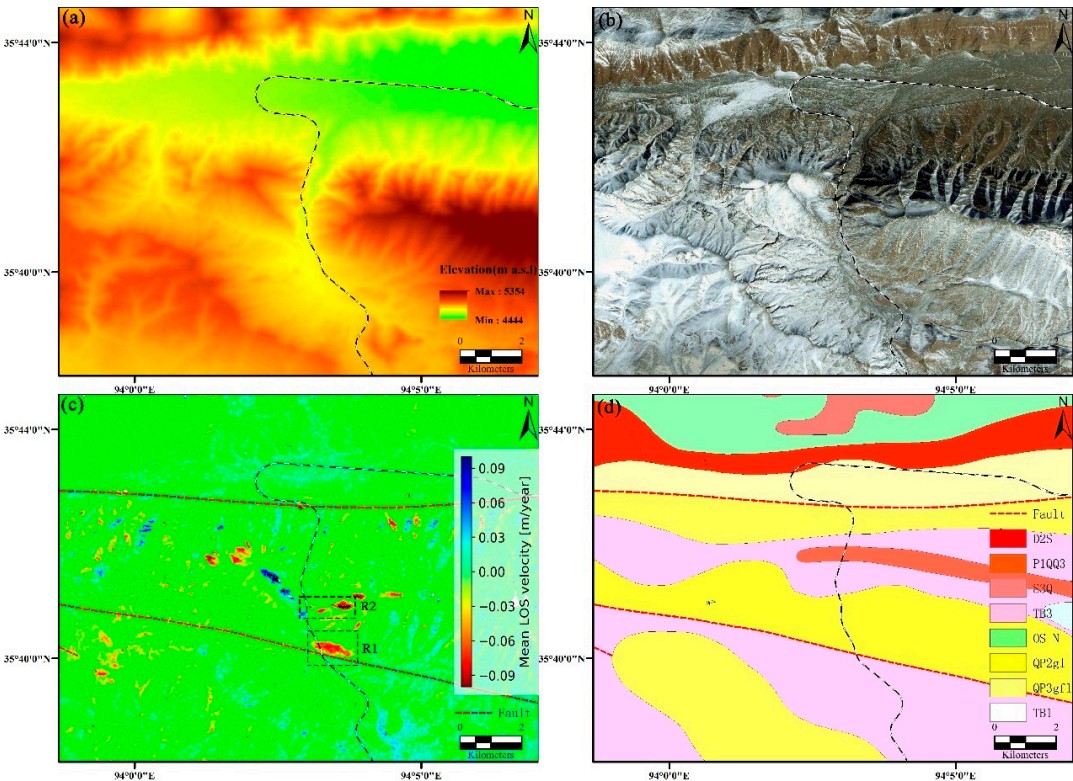

**Figure 6.** (**a**) shows the DEM of the railway section; (**b**) is the optical image of Google Earth; (**c**) is the distribution of surface deformation at unfrozen spring station; (**d**) is a geological distribution map, and the fault zone is shown in red dotted line.

*4.2. Comparison and Discussion on Deformation Law of Road Section*

4.2.1. Comparison of Different Deformation of Roadbed Frozen Soil

In order to compare the different time series deformation laws of WB and TZ deformation sections and understand their differences in climate response, we obtain the surface time series deformation information from the deformation risk area, as shown in Figure 8. At the same time, in order to more accurately judge the response of time series deformation and climate change, we take the actual distance from each risk area to the meteorological station as the weight factor to obtain the weighted average temperature and precipitation.

In addition to the thawing and collapse of frozen soil in WB section, at the west side of the railway passing through Kunlun Mountain, the central latitude and longitude are 35.67944° N and 94.04924° E. The roadbed has been raised due to frost heaving of some frozen soil, with cumulative deformation of 50.507 mm and annual average deformation of 20.09 mm/a. From January to early March, the ground surface is constantly lifted due to the frost heaving of frozen soil. In summer, the frost heaving slows down due to the increase in temperature and precipitation, which is basically in a constant trend. In autumn and winter, when the temperature decreases, the surface temperature also decreases. The water in the active layer of frozen soil under the surface solidifies, and the frozen soil heaves. The maximum value of frozen soil heave is rising, as shown in Figure 8c. This shows that the active layer thickens, and because it is located on the shady slope, the frozen soil layer has good development conditions, which belongs to the developing permafrost.

The settlement area of TZ section is rapidly affected by the climate. From the end of April, the precipitation and temperature began to rise, the frozen soil active layer gradually began to melt, and the surface settlement is serious. Its trend line is shown in Figure 8d. The annual average time series deformation rate is 38.47 mm/a, the cumulative deformation variable is 86.39 mm, and the latitude and longitude of the deformation center are 32.8847° N and 91.5283° E. The seasonality of the climate is seriously affected by the

southeast monsoon. Most of the precipitation occurs from June to August in the form of rainstorm, which leads to flash floods and extensive surface erosion. At the same time, it also leads to the intensification of the melting of frozen soil active layer. The remaining precipitation occurs in the form of snow or hail. Sometimes snow will be generated during the ice period (usually 7–8 months, from September to April of the next year), which is consistent with the deformation. From September to April of the next year, the deformation will obviously slow down, while from June to August, the frozen soil will continue to melt under the influence of temperature and precipitation, resulting in intensified surface settlement. As the settlement area is located in the roadbed section of Qinghai–Tibet railway, it poses a threat to railway operation.

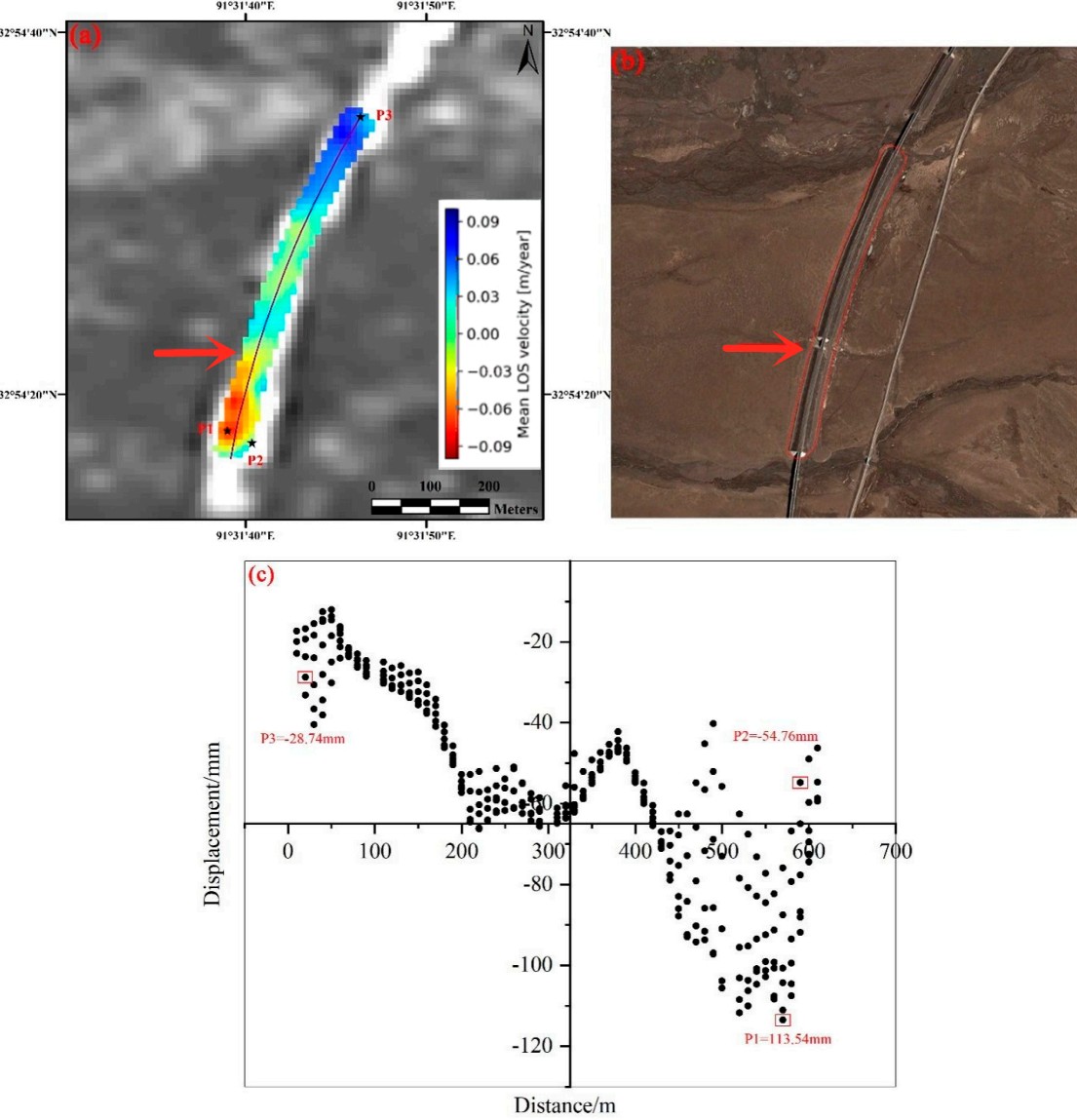

**Figure 7.** (**a**–**c**), respectively show the annual average deformation, topographic and temporal deformation of railway roadbed. The red arrows in (**a**,**b**) show the runoff formed by the thermal melting of frozen soil and the boundary of different deformation mechanisms at the north and south ends. The red box indicates the roadbed range of the railway section. The abscissa in (**c**) represents the distance of p3–p1 point from north to south, and the ordinate is the deformation.

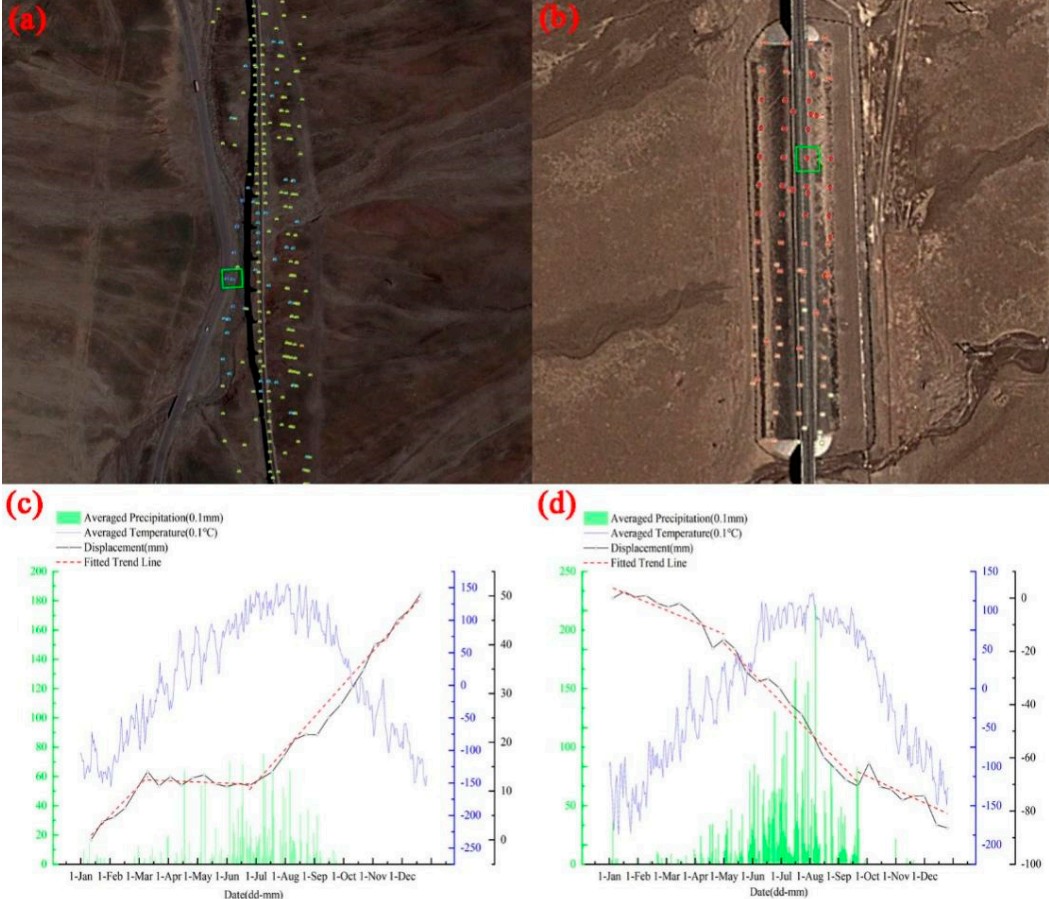

**Figure 8.** (**a**,**b**) depict the distribution of roadbed deformation in WB and TZ sections, respectively, (**c**,**d**) depict the time series deformation of railway roadbed in WB and TZ sections under the influence of climate factors, respectively. The green box in the optical diagram is the selection point of time series deformation, and the red dotted line in the broken line diagram is the deformation subsection fitting line.

### 4.2.2. Comparison of the Latest Available Results with Previous Results

Because Beilu River and Tuotuo River are located in the basin area, the river system has a great impact on the frozen soil, and the frozen soil section often becomes a research hotspot. This paper compares the deformation of the reach from Wudaoliang to Tuotuo with the study of Zhang et al. from 2009 to 2018, as shown in Figure 9. The three places C1, C2 and C3 pass through Beiluhe basin, Fenghuoshan area and Tuotuohe basin, respectively. C1 and C3 are located in the valley basin, rich in water resources and geographical environment, and there is no high mountain shelter. The frozen soil is greatly affected by thermal melting and forms a large number of thermal melting lakes, as shown in Figure 10a,b. Therefore, these two areas are scattered point deformation areas in InSAR detection results. There is no obvious frozen soil collapse at the railway roadbed. However, under the global warming environment and the expansion of inland rivers and thermal melting lakes, the frozen soil under the railway roadbed will also be affected, and there is still a certain risk of thawing collapse. The deformation at C2 is small, and it has been significantly improved compared with the results in 2018. This area not only takes heat dissipation measures for frozen soil, as shown in Figure 10c, but is also located in high mountain and valley area (Figure 10d), with an altitude of 5000–5200 m and a height drop of 400–600 m with the railway section. The risk of frozen soil collapse is further reduced. In short, although there is no serious deformation of railway roadbed at C1, C2 and C3, considering that there are still many thermal melting lakes near C1 and C3, it is still necessary to monitor the frozen soil section.

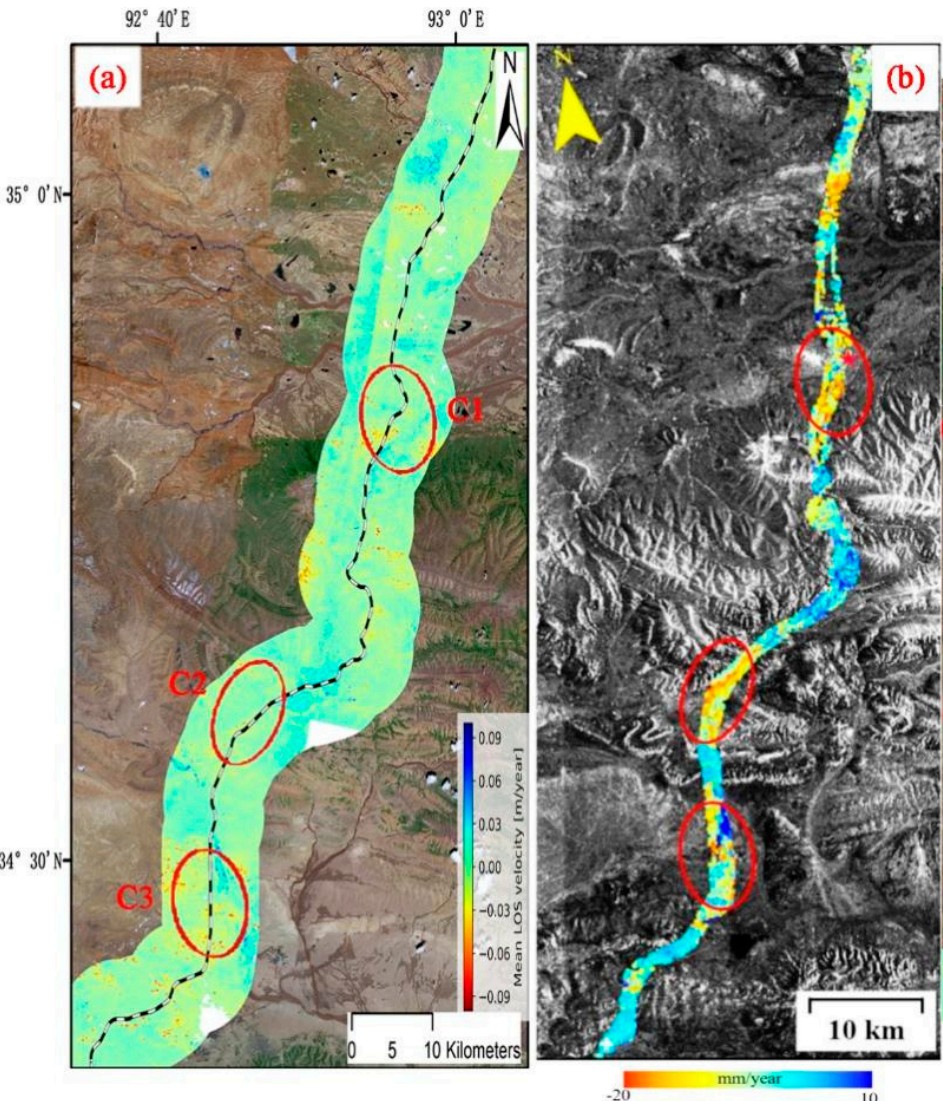

**Figure 9.** (**a**) shows the annual average time series deformation of the railway corridor from Beilu River to Tuotuo River in 2020, and (**b**) shows the deformation detection results of Zhang et al. in 2018. The three deformations detected by Zhang are marked with circles in the Figure, namely C1, C2 and C3. The deformation area obviously decreases, especially at C2. No obvious deformation is detected at C1 and C3 railway roadbed.

*4.3. Uncertainty Analysis of Results*

Due to the destructive effects of atmospheric precipitation (especially snowfall), the formal application of InSAR technique to monitor structures generating scattering may provide incorrect results. The corrupting impact of atmospheric precipitation on the phase of reference targets has been mentioned in many studies. In winter, the main source of alteration for the propagating signal properties is the growth in snow depth between the SAR observations [67].

As shown in Figure 10, part of our study area is covered by snow because it is located in the Qinghai–Tibet plateau. Snow can affect radar interference and skew results. Therefore, in addition to removing atmospheric phase, terrain phase and noise, we also need to remove the influence of snow removal. As the influence of snow in the main deformation area is small, this study does not deal with the influence of snow temporarily. However, the impact of snow removal will be the focus of our next study.

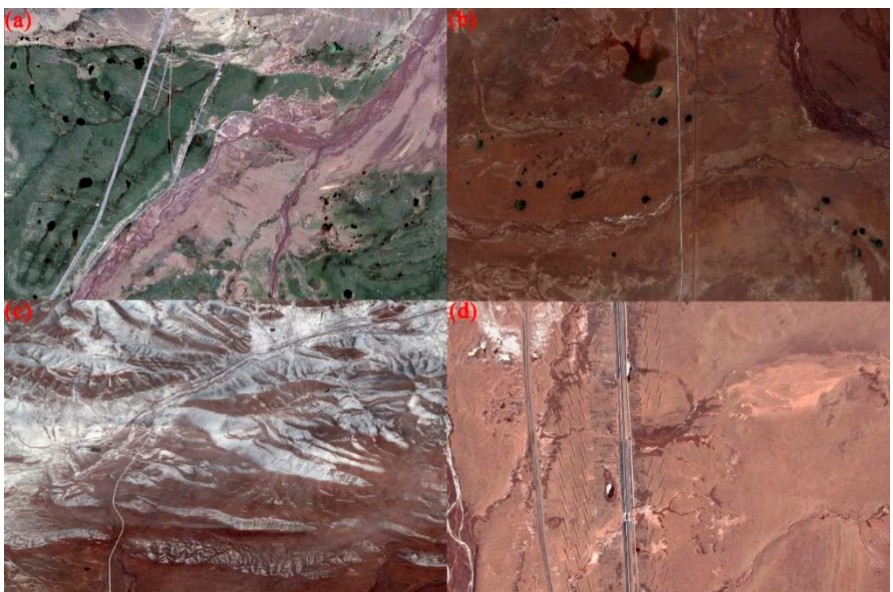

**Figure 10.** In (**a**,**b**), there are widely distributed hot melt lakes formed by ground collapse after frozen soil melting. (**c**,**d**) are the topographic distribution and solutions of Fenghuo Mountain section, respectively. It can be seen that hot rods are arranged on the left and right sides of the road to dissipate heat and reduce the impact of frozen soil hot melt and frost heaving on the roadbed.

## 5. Conclusions

In this study, four Sentinel-1 satellite maps are selected to fully cover the study area, with a total of 122 images. The MT-InSAR technology is carried out for the 610 km-long Qinghai–Tibet railway section (from Naij Tal railway station to Anduo railway station) in the permafrost area so as to obtain the time series deformation information of the surface along the Qinghai–Tibet railway and compare its deformation with climate factors. The main conclusions are as follows:

(1) The areas with serious deformation of the Qinghai–Tibet corridor are mainly distributed in the railway section from WangKun station to Budongquan station and the section from Tanggula station to Za'gya Zangbo station, and there are many areas of railway roadbed subsidence and mountain collapse.

(2) The influence of the frozen soil section from WangKun station to Budongquan station on the railway roadbed is high in the middle and low at both ends. The influence of human activities of railway operation on the frozen soil deformation is smaller than that of topography and hydrothermal. At the same time, the geological strata and fault zone of this section also have a certain impact on the roadbed deformation. D2S, OS, and S3Q frozen soil layers are more stable than Qp2gl and Qp3gl strata.

(3) Between the Tanggula and Za'gya Zangbo station, there was a 620 m-long railway roadbed with uneven deformation on the east and west sides, with an average annual difference of 60.68 mm/a. At the same time, uneven deformation also exists in the railway roadbed on the north and south sides.

(4) Through comparison, it is determined that the roadbed deformation does not exist in the area from Beilu River to Tuotuo River, and the permafrost has no great impact on the railway roadbed.

(5) Under the situation of global warming, the frozen soil will continue to undergo thermal thawing and frost heaving. At the same time, the permafrost will continue to decrease and the active layer will continue to thicken. In order to ensure the stable operation of the Qinghai–Tibet railway in the permafrost section, it is necessary to regularly monitor the deformation of the permafrost area up to 610 km by MT-InSAR.

At the same time, our research has some shortcomings. The impact of snow cover on InSAR interference in the study area needs further treatment. This is the focus of our next research work.

**Author Contributions:** H.L. and S.H. performed experiments, analyzed the data and prepared the manuscript. C.X. and B.T. provided crucial guidance and support through the research. M.C. significantly contributed to the validation work and data interpretation. Z.C. provided valuable suggestions for this study. All authors have read and agreed to the published version of the manuscript.

**Funding:** This research was funded by Fujian Provincial Science and Technology Project (Science and Technology Service Network Initiative, CAS) (2020T3011), jointly funded by the Outstanding Youth Science and Technology Program of Guizhou Province of China ([2021]5615).

**Data Availability Statement:** The data supporting the findings of this study are available from the first author (H.L.) upon reasonable request.

**Acknowledgments:** The authors would like to thank ESA (European Space Agency) and Alaska satellite facility (https://asf.alaska.edu/, accessed on 30 July 2021) for providing the Sentinel-1 datasets of the Copernicus mission.

**Conflicts of Interest:** The authors declare no conflict of interest.

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
