# Peer review of "Monitoring Roadbed Stability in Permafrost Area of Qinghai–Tibet Railway by MT-InSAR Technology"

_land, doi:10.3390/land12020474_

Round 1
Reviewer 1 Report
I think this paper can be accepted as it is.
Author Response
Thank you very much for your acknowledgement of our manuscript.
Reviewer 2 Report
The paper Monitoring roadbed stability in permafrost area of Qinghai-Tibet railway by MT-InSAR Technology (Hui Liu, Songbo Huang, Chou Xie, Bangsen Tian, Mi Chen and Zhanqiang Chang) is devoted to topical issues related to the safe operation of railways in permafrost. In such regions, permafrost effects are possible due to changes in the structure of the soil (frost heaving of the soil, thawing and freezing of soils). These effects lead to deformation of the railway. The authors studied this phenomenon on the example of the Qinghai-Tibet Railway using InSAR time series technology to monitor such deformations. This technology makes it possible to obtain information on ground deformation with accuracy of up to millimeters. The authors have identified specific sections of this railway where the maximum deformation is observed. At the same time, the main characteristics that have the maximum impact on these processes are the influence of topography and hydrothermal conditions.
Authors are encouraged to improve the English language of the article and correct errors, including punctuation.
For example,
-l 59 – Need…landslides, and….
-l 92- montioring Or monitoring?
-l 101- make it Or makes it?
-l 204 Remove spaces in expression ∆ y (m)
-l 292- . a 'are necessary a' are
-l 337; -l 404 Need …., and
-l 342 from Tanggula station to zajiazangbu Or from Tanggula station to Zajiazangbu?
-l 377; -l 453 Figure.s Or Figures? Bad point notation: In Figure.s a and b; soil melting. c and d
- l 504-l 513 Appendix A, B? What for?
Unsuccessfully formatted links throughout the article: no spaces.
For example,
-l 38 ….highway[3-5] Need …highway [3-5]
Some difficulties in reading the article are caused by the abbreviations, which it is desirable to put into a separate appendix. It would be useful to recommend the authors to issue an appendix with a list of abbreviations.
Perhaps it would be useful for the authors of the article to publish this article in a special issue of the journal LAND journal dedicated to permafrost problems. For example, in Special Issue Title: Forecasting of Permafrost Boundaries Dynamics
https://www.mdpi.com/journal/land/special_issues/permafrost_dynamics
After the corrections the paper can be published in the journal LAND.
Author Response
We gratefully thank editor and all reviewers for their time spend making their constructive remarks and useful suggestions, which has significantly raised the quality of the manuscript and has enables us to improve the manuscript. Each suggested revision and comment, brought forward by the reviewers was accurately incorporated and considered. Below the comments of the reviewers are response point by point and the revisions are indicated. And the full text has been revised carefully for another time.
The specific modification information is in the PDF we uploaded.

Reviewer 3 Report
The problem of deformations due to seasonal effects of freezing and thawing is significant for many regions of the planet. However, only one region is indicated in the introduction. This aspect should be expanded. Besides the problem should be considered more broadly, i.e., examples of using MT-InSAR technology for various applications should be indicated.
Line 159-161 - The meaning of the sentence is not quite clear, it needs to be reformulated.
Line 352 - It is unclear about which arrow. Perhaps it is necessary to rearrange the first few sentences (325-355) and add explanations.
It is necessary to add a color scale for Fig. 7.
Winter and summer data of Sentinel-1 were used, and how was the impact of snow taken into account? In the optical image of Fig. 6, I see snow. The effect of snow should be assessed for example see
Author Response

(The authors gave the same response as above.)

Round 2
Reviewer 3 Report
Only part of the comments have been taken into account. The main comments have been ignored, so I have to repeat my comments:
- The problem of deformations due to seasonal effects of freezing and thawing is significant for many regions of the planet. However, only one region is indicated in the introduction. This aspect should be expanded. Besides the problem should be considered more broadly, i.e., examples of using MT-InSAR technology for various applications should be indicated.
- Winter and summer data of Sentinel-1 were used, and how was the impact of snow taken into account? In the optical image of Fig. 6, I see snow. The effect of snow should be assessed for this investigation. For example see:
Yichen Yang, Shifeng Fang, Hua Wu, Jiaqiang Du, Xiaohu Wang, Rensheng Chen, Yongqiang Liu, Hao Wang. (2022) High-Resolution Inversion Method for the Snow Water Equivalent Based on the GF-3 Satellite and Optimized EQeau Model. Remote Sensing 14:19, pages 4931. Adebisi Naheem Idowu, Hans-Peter Marshall. (2022) Snow depth retrieval from L-band data based on repeat pass InSAR techniques. IGARSS 2022 - 2022 IEEE International Geoscience and Remote Sensing Symposium, pages 4248-4251. P.N. Dagurov, T.N. Chimitdorzhiev, A.V. Dmitriev & S.I. Dobrynin (2020) Estimation of snow water equivalent from L-band radar interferometry: simulation and experiment, International Journal of Remote Sensing, 41:24, 9328-9359, DOI: 10.1080/01431161.2020.1798551
Author Response
Many thanks to the reviewers for their valuable comments. We have modified the manuscript accordingly, and the detailed modification information is in the PDF file we uploaded.

Round 3
Reviewer 3 Report
Once again, I ask you to evaluate the impact of snow, for example, how it is done in these works:
Zakharov, A.; Zakharova, L. An Influence of Snow Covers on the Radar Interferometry Observations of Industrial Infrastructure: Norilsk Thermal Power Plant Case. Remote Sens. 2023, 15, 654. https://doi.org/10.3390/rs15030654
Author Response
Thank you very much for your comments. The literature you provided is very valuable. Snow does exist in our study area, but its effect on our study is very small. In other studies on permafrost monitoring, there is also less discussion on snow. Your opinion will be further studied in our subsequent work, but snow is not the focus of our study area. Thank you again for your valuable advice.